



# Present-climate trends and variability in thermohaline properties of the northern Adriatic shelf

Ivica Vilibić[1*], Petra Zemunik[1], Jadranka Šepić[1], Natalija Dunić[1], Oussama Marzouk[2], Hrvoje Mihanović[1], Clea Denamiel[1], Robert Precali[3], Tamara Djakovac[3]

[1] Institute of Oceanography and Fisheries, Split, Croatia
[2] student at SeaTech, University of Toulon, Toulon, France
[3] Center for Marine Research, Ruđer Bošković Institute, Rovinj, Croatia

Correspondence to: I. Vilibić, vilibic@izor.hr

**Abstract:** The paper documents seasonality, interannual to decadal variability and trends in temperature, salinity and density over a transect in the shallow northern Adriatic Sea (Mediterranean Sea) between 1979 and 2017. Amplitude of seasonality decreases with depth, and is much larger in temperature and density than in salinity. Interannual to decadal variability in temperature and salinity are differently correlated in surface and bottom layers, indicating different mechanisms which govern their variability. Trends in temperature are large (up to 6ºC over 100 years), significant through the area and not sensitive to

the sampling interval and time series length. In contrast, trends in salinity are largely weak and insignificant and depend on the time series length. The warming of the area is stronger during spring and summer. Such large temperature trends and their spatial variability indicate substantial changes in the thermohaline circulation in this area known as a dense water formation site, with a potential to affect biogeochemical and ecological properties of the whole Adriatic Sea.

## 1 Introduction

20       Albeit being shallow, with depths lower than 80 m and of limited dimensions (ca. 150 km x 250 km), the northern Adriatic shelf (Fig. 1) has been early recognized as an important area in the Mediterranean, as a number of important ocean processes occur there. These include: (1) tides, which are strongly amplified due to their near-resonance with the Adriatic eigenoscillations and surpass 1 m at the northernmost part of the Adriatic (Janeković and Kuzmić, 2005; Vilibić et al., 2017), (2) storm surges, which are generated by a strong and persistent sirocco wind and which flood coastal cities in the northern

Adriatic, like Venice (Trigo and Davies, 2002; Međugorac et al., 2018), (3) a substantial freshwater discharge (Raicich, 1996), that drives the surface branch of the Adriatic Sea thermohaline circulation, outflowing along the western coast and causing the counterflow along the eastern coastline that brings saline Levantine Intermediate Water (LIW) all the way to the northern Adriatic (Orlić et al., 1992; Artegiani et al., 1997), and (4) dense water formation (Bergamasco et al., 1999; Beg Paklar et al., 2001; Mihanović et al., 2013), which occurs during the wintertime cold bora outbreaks at the shelf (Grisogono and Belušić,



2009), resulting in generation of the densest Mediterranean water mass, the North Adriatic Dense Water (NAdDW, Zore-Armanda, 1963). The NAdDW spreads over the deep Adriatic layers as a density current (Artegiani and Salusti, 1987) and ventilates the middle and southern Adriatic depressions (Vilibić, 2003; Bensi et al., 2013).

All of the quoted processes are, to certain extent, influenced by climate changes, thus a prerequisite for proper assessment of their changes is a long-term monitoring. Due to their quasi-resonant nature, the tides are sensitive to changes in mean sea level, i.e. the sea level rise of 2 m is going to increase major diurnal tide $K_1$ for 25% and decrease major semidiurnal tide $M_2$ for 10% in the northern Adriatic (Lionello et al., 2005). The Adriatic storm surges are sensitive to the intensity and frequency of cyclones, projected to decrease in their frequency but not in peak intensity in the future climate (Androulidakis et al., 2015; Lionello et al., 2017b). Also, a small change of synoptic patterns may change distribution of storm surge heights along the coastal northern Adriatic (Međugorac et al., 2018). A pronounced warming trends with maximum in the summer season and a substantial decrease in precipitation, especially in the warm season (Giorgi, 2006; Giorgi and Lionello, 2008; Planton et al., 2012; Branković et al., 2013; Lionello and Scarascia, 2018), are going to affect river discharges in the Adriatic. Yet, the Alpine region is projected to be less influenced by the precipitation decrease, having even an increase in its northern parts (Gao et al., 2006), as a dividing line between the Mediterranean drying regime and the continental wetting regime in precipitation projections is stretching along the northern Adriatic drainage areas (Zampieri et al., 2012). For that reason, freshwater influx to the Adriatic Sea through the northern Adriatic rivers will change with climate, but more regarding seasonality than total runoff (Coppola et al., 2014). Last but not least, dense water formation in the Adriatic is found to have a decreasing trend in the present climate, at least as seen on the thermohaline circulation reproduced by ocean climate simulations (Somot et al., 2006) and long-term measurements along the Palagruža Sill transect (Vilibić et al., 2013).

The northern Adriatic Sea is one of rare Mediterranean regions where a tradition of continuous high-resolution (monthly or bimonthly) monitoring of oceanographic parameters has been preserved over a prescribed set of stations. Such an approach is sparse in the rest of the Mediterranean (Malanotte-Rizzoli et al., 2014). These measurements allowed for documentation of the functioning of the northern Adriatic ecosystem over long-term (Plavšić, 2004; Mozetič et al., 2010; Ivančić et al., 2010; Kraus and Supić, 2011; Marić et al., 2012; Giani et al., 2012; Djakovac et al., 2015; Iveša et al., 2016; Dautović et al., 2017). Previous works also include analysis of variability and changes of the thermohaline properties (Supić et al., 2004) and their drivers, freshwater discharges and air-sea fluxes (Supić and Ivančić, 2002). The long-term trends in temperature and salinity over a centurial timescale have not been found significant in the eastern part of the northern Adriatic shelf (1921-2000, station RV001, Supić et al., 2004), although the deep southern Adriatic exhibited a centenarian warming (1911-2009, Lipitzer et al., 2014). Yet, the temperature and salinity trends in the last few decades have been found significant in different parts or the whole Mediterranean Sea (Shaltout and Omstedt, 2014; Grbec et al., 2018; Pastor et al., 2018; Bengil and Mavruk, 2019). This applies both to the surface, but also to the deep layers to which warming induced by climate change propagates more slowly (Bethoux et al., 1990; Tsimplis and Baker, 2000; Millot et al., 2006; Cusinato et al., 2018).

The Adriatic Sea *in situ* ocean observations are continuous at the Palagruža Sill transect in the middle Adriatic as well. The Palagruža series have been exploited in a number of studies (i.e. Grbec et al., 2009; Vilibić et al., 2012, 2013;



Mihanović et al., 2015), as this transect is located at the key region for water mass exchange between the deep and the shallow Adriatic and Mediterranean regions (Buljan and Zore-Armanda, 1976; Martin et al., 2009). Therein trends indicate a weakening of the Adriatic-Ionian thermohaline circulation (Vilibić et al., 2013), which is determined by the dense water formation and freshwater discharges in the northern Adriatic, and inflow of the LIW, and dominant circulation patterns in the Ionian. Yet,

the northern Adriatic thermohaline measurements, in particular their variability and trends, have not been analyzed during the recent decades, when accelerated changes in the Mediterranean climate have been found to occur (Lionello et al., 2017a). Our study bridges this research gap, analyzing thermohaline variability and trends during the ERA-Interim period (1979-2017). The emphasis is particularly given to the present-climate trends and their persistence. Section 2 presents the data and the methods. An analysis of seasonal variance, interannual variability and trends are given in Section 3. Section 4 discusses the

results and highlights the major findings.

## 2 Data and methods

*In situ* temperature and salinity data were collected at six stations surveyed mostly monthly or bimonthly between 1979 and 2017: RV001, SJ107, SJ105, SJ103, SJ101 and SJ108 (Fig. 1, in order from Croatian to Italian coastline). The samples were taken at 0, 5, 10, 20, 30 m and 2 m above the seabed. Temperature was measured by protected reversing

thermometers (Richter and Wiese, Berlin, precision ±0.01 °C) and by reversing digital thermometers (SIS RTM 4002, precision ±0.003 °C) assembled to the Niskin bottles. Salinity was determined using Mohr and Knudsen's method with an accuracy of ±0.05 (data between 1966 and 1977) or by high precision laboratory salinometers with accuracy of ±0.01 (data from 1978 onwards). For months with two or more cruises, the averages were used in analyses. The number of samples in a month varied between 19 to 25 in November and 37 to 40 in July, with sampling more frequent during summer season and in March than

during the rest of a year (Fig. 2).

All data were checked for quality by using min–max thresholds (4-32°C for temperature, 10-40 for salinity) defined by long-term climatology of the northern Adriatic (Artegiani et al., 1997; Lipizer et al., 2014) and by checking if vertical stability of the water column is preserved. Then, visual checking and comparison of each data to its neighbors in both space (measurements at neighboring depths and stations) and time were carried out. Potential density anomalies (PDAs) with a

reference pressure of zero were estimated using TEOS-10 algorithms.

Seasonal changes of all variables were largely removed from each series by fitting annual (12 months) and semi-annual (6 months) cosine functions to data. Such an approach allowed for larger significance of trend estimates and proper statistics (the estimate of annual averages by taking into account uneven samplings during different seasons). The procedure was separately applied for each station, depth and variable. The remaining signal within a time series, i.e. without annual and

semi-annual cycle (referred as seasonal series), is referred as residual series in the paper.

The trend significance was estimated by Mann-Kendall nonparametric test.



## 3 Results

### 3.1 Averages, seasonal cycle and variance

The averages of temperature, salinity and PDA estimated from the residual series (Fig. 3) exhibit a substantial decrease in temperature with depth. This is the consequence of (i) cold and dense waters generated during a winter, residing

at the bottom and increasing stratification of the water column through rest of the year, and (ii) a flooding of the most of the northern Adriatic by waters of low salinity during spring when thermocline is formed, increasing again stability of the water column. These two effects keep heat accumulated in the surface layer and prohibit the ventilation of the water column till November, when the vertical mixing prevails to the buoyancy (Supić et al., 2004). The flooding of the whole Rovinj-Po transect with low-salinity waters coming from the northern Adriatic rivers is reflected in the mean salinity distribution, with average

salinity lower for about 4.0 at the surface than at the bottom off the Po River mouth (station SJ108), while surface salinity is lower on the western section (33.7 in average at SJ108) than on the eastern section (37.3 in average at RV001) of the transect. The maximum in salinity is documented in near-bottom layers of stations SJ107 through SJ105, indicating the area where the largest advection of saline waters from the southeast is occurring. PDA dominantly reflecting effects of salinity in the surface layer and both temperature and salinity in the bottom layer, where the densest waters are kept at the bottom of the central and

western sections of the transect (stations SJ105 to SJ101).

Seasonal changes in the surface layer are quite strong (Fig. 4), with surface temperature reaching minimum of 8-9ºC in February and maximum of 25-26ºC in August. Monthly max-min range in the surface temperature is slightly larger on the western (SJ101) than on the eastern (RV001) section of the transect, due to stronger haline-driven stratification that decreases the rate of vertical heat exchange between surface and bottom layers. For the same reason, monthly max-min range in the

bottom temperature is lower on the western than on the eastern section of the transect. Surface salinity is largest during wintertime (37.3 at SJ101 in January and 38.0 at RV001 in February), when the freshening by rivers is restricted to the western coastline (Artegiani et al., 1997), while it falls down to 32.3 in May and 36.3 in July at SJ101 and RV001, respectively, following the spring maximum in river discharges (Raicich, 1996). Bottom salinity does not change much, having values around 38.0 throughout a year. PDA seasonal cycle is affected more by temperature changes at the eastern section of the

transect (up to 70% of PDA surface change is due to temperature changes), whilst it is influenced more by salinity seasonal changes at the western section of the transect (up to 55% of PDA surface change is due to salinity changes). The largest PDA values are measured in February, with near-bottom values between 29.4 and 29.5 kg/m$^3$ along the transect. These numbers indicate generation of the NAdDW over most of the northern Adriatic, filling the northern Adriatic shelf before flowing to the southeast along the western shelf break (Artegiani and Salusti, 1987; Janeković et al., 2014).

The amplitude of changes of the temperature seasonal series is much larger than of the temperature residual series, in both surface and bottom layers (Fig. 5). The same applies to PDA, while the annual cycle of salinity has much lower amplitudes, particularly in the bottom layer. The dominance of the seasonal signal in the overall temperature changes is emerging from percentages of annual and semi-annual cycle variance versus total variance (Fig. 6), which is above 90% at the



surface and is not lower than 75% near the bottom. Variance of seasonal series in salinity surpass 30% at the surface of the eastern section of the transect, and 20% at the surface of the western section of the transect (station SJ108). This implies that transport of the Po River waters towards the eastern coastline has a larger seasonality, whilst station SJ108 is affected by the river plume uniformly throughout a year (Kourafalou, 1999). In the bottom layer, variance of salinity seasonal series is low
(<5%), indicating a dominance of processes on interannual to decadal over seasonal timescale, as well transient changes occurring over a few months period. The PDA seasonal variance is affected by both temperature and salinity, being largest in the subsurface layer on the eastern section of the transect (80-90%), while being between 60 and 70% at the bottom of the western section of the transect.

### 3.2 Interannual variability

Hovmoller plots of residual temperatures, salinity and PDA series (Fig. 7) show a strong interannual signal, both in surface and bottom layers. Temperature and salinity at 0 m are correlated at 95% at all but SJ108 stations, with an increase in temperature associated with a decrease in salinity and vice versa. Yet, temperature and salinity are not correlated at 95% in the bottom layer at any but RV001 station, indicating the dominance of different mechanisms which drive the bottom temperature and salinity variability. The surface variability is presumably dominantly a result of the direct forcing from the atmosphere
associated with the hemispheric patterns that are known to influence Central Mediterranean temperatures and precipitation (like East Atlantic pattern, Ionita et al., 2015; Scorzini et al., 2018). The bottom variability is largely resembling an advection of salt and heat from the southeast through the Adriatic-Ionian thermohaline circulation (Orlić et al., 2006). An overall increase in temperature at all stations can be visually detected, with prolonged cooler periods in the bottom layer at the beginning of the series (e.g. 1983-1987), and shorter periods of above-average temperatures more frequent in the second part of the series
(e.g. 2000-2001, 2007, 2014).

Regarding residual salinity series, it seems to be a combination of interannual (1-3 years) and quasi-decadal variability (5-10 years). There can be noticed prolonged periods of higher salinity (e.g. 1987-1991, 2002-2008), interrupted with shorter periods of lower salinity (e.g. 1984-1986, 1991-1998, 2009-2015), on which interannual variability is superimposed. It is a question if this variability is connected with the oscillatory Adriatic-Ionian Bimodal Oscillating System (BiOS, Gačić et al.,
2010), which has been found to affect southern and middle Adriatic (Civitarese et al., 2010; Mihanović et al., 2015). The BiOS was in anticyclonic regime between 1991 and 1998 (low salinity conditions), then switching to cyclone regime between 1998 and 2006 (high-salinity conditions), again to anticyclone regime till 2011 and being in cyclone regime till recent years (Fig. 7, Gačić et al., 2010, 2014; Mihanović et al., 2015). It looks that high and low salinity periods in the northern Adriatic are lagging a few years after the BiOS reversals. Yet, quantification of this connection is not the objective of this paper, and will be
investigated in future research, together with other potential drivers of the thermohaline variability in the Adriatic Sea.

Residual PDA interannual to decadal changes are dominantly affected by salinity (ca. 80-90% and ca. 55% of residual PDA surface and bottom change, respectively, are due to residual salinity changes), having the range in near-bottom values (30 m) between 27.2 kg/m$^3$ at station SJ103 and 29.7 kg/m$^3$ at station SJ105. Minimum densities are reached in periods 1993-



1995, 2000-2002 and 2007-2015, while highest residual PDA values are observed in period 2003-2005. The last period matches the ending years of the strong cyclonic BiOS regime (1998-2006, Mihanović et al., 2015). Similarly, low PDA values lag for a few years for the onset of the anticyclonic BiOS regimes.

### 3.3 Trends

Temperature, salinity and PDA trends estimated on residual annual averages, and presented in Fig. 8, reveal an extensive and statistically significant heating of the whole water column over the whole transect. Trends range from 1-6°C over 100 years, reaching maximum value off the Po River at the bottom of the SJ108 station. Salinity and PDA trends at this station are also large, indicating a weakening of stratification in recent time south of the Po River delta, and implying that this higher-than-average temperature bottom trends might be due to an increase of heat transfer towards the bottom in this area.

Temperature trends are lowest on the eastern section of the transect (1-2°C over 100 years), where an inflow of waters from the middle Adriatic normally occurs (Franco et al., 1992; Orlić et al., 1992; Artegiani et al., 1997).

Salinity annual trends are mostly insignificant over the transect. Significant increase in salinity is documented at the surface of station SJ107, indicating a mild increase of salinity along the eastern part of the section. The increase is presumably the result of a combination of persistent salting of the middle and southern Adriatic (Vilibić et al., 2013; Lipizer et al., 2014)

advected towards the northern Adriatic and effects of freshwater influx from the northern Adriatic rivers. Interesting surface salinity trends are found off the Po River, where the trend at SJ101 is negative while strongly positive at SJ108 (>6 over 100 years). As these stations are occasionally flooded by the Po River plume (Kourafalou, 1999), the trends indicate a change in the Po River dynamics, i.e. that the plume has been spreading more towards the east during the recent decades, while it spreads more to the south in the 1980s and 1990s.

The bipolar surface structure off the Po River is also detectable from the PDA trends, while trends over the rest of the section are both thermally- and haline-driven. For that reason, the PDA trends are mostly negative over the transect, with rates from -3 kg/m$^3$ to -1 kg/m$^3$ over 100 years.

Trends differ between months. January temperature trends (Fig. 9) are positive, with maximum in bottom layers, at central part of the transect and at the station SJ108 (> 4°C over 100 years). Salinity trend follows temperature trend in the

central section of the transect, indicating an increased rate of advection of the middle Adriatic waters to the northern Adriatic. Trends in summer months, in particular July, have much more complex spatial structure due to baroclinicity. Temperature trends surpass 6°C over 100 years at the very surface and off the Po River delta, while they are insignificant and even opposite at the bottom of central and eastern parts of the transect. Such a distribution indicates that vertical transfer of heat has been reducing in the central and eastern sections of the transect, resulting in accumulation of the heat energy close to the surface. In

contrast, temperature trends are largest at the bottom of SJ108 station, where larger heat transfer to the bottom is presumably allowed by increased surface salinity and therefore reduced vertical stratification. Salinity trend in July is positive in the western part of the section, peaking at the station SJ108 (8.6 over 100 years).



Seasonal changes in trends (Fig. 10) reveal that trends are more intense and more significant during spring and summer. Largest temperature trends on the eastern side of the transect (RV001) are documented between April and August: in the bottom layer from April through May and in the surface layer between July and August. The bottom trends go to near-zero values between July and November, revealing stability of weak vertical exchange of heat due to increased stratification and

lowered mixing during late summer/autumn. These trends also reflect trends in air temperature which are larger during the summer and lower during the winter in the region (Shohami et al., 2011; Bartolini et al., 2012). A secondary maximum in temperature trends at the eastern side of the transect occurs in December. Temperature trends at SJ101 are similar to trends at RV001, but even higher during spring and summer (occasionally >8°C over 100 years), reflecting coherent seasonal changes over most of the transect. Yet, temperature trends at the station SJ108 are different, with spring and summer maximum

restricted to depths higher than 10 m.

Salinity trends differ more lot between the eastern and the western parts of the transect. Trends are insignificant at the eastern part of the transect, weakly negative in spring and positive in summer (only in surface layers) and in December. In contrast, salinity trends are strongly negative at the surface of the SJ101 station during most of the year (Fig. 10, <-4 over 100 years), indicating positive trends in offshore expansion of the Po River plume. Simultaneously, salinity trends are strongly

positive at the surface of the SJ108 station (occasionally >7 over 100 years). Therefore, the change of the Po River plume dynamics and of the pathway of freshwater in the northern Adriatic in recent decades was substantial and persistent in all seasons.

PDA trends are negative at most of the transect in all seasons, except at surface layers of the SJ108 station. Decrease in PDA, although not significant, is also evident in the wintertime period (January-February), during which the NAdDW is

generated in the northern Adriatic (Zore-Armanda, 1963; Artegiani et al., 1997; Mihanović et al., 2013). The 30-m February PDA trends range between -1.1 kg/m$^3$ at SJ101 and -0.8 kg/m$^3$ at RV001. The period in which the NAdDW spread to the middle and southern Adriatic, March-June, is characterized with even more negative PDA trends, driven by both temperature and salinity trends. A negative salinity trend indicates a weakening of advection of saline middle Adriatic waters to the northern Adriatic, due to a weakening of the Adriatic thermohaline circulation (Vilibić et al., 2013).

As the series are relatively short, it is a question if the trends are sensitive to the choice of the sampling interval or have a persistence regardless of the sampling interval. For temperature, trends are persistent (Fig. 11), significant and change a little over the whole transect, regardless of inclusion of recent data in the analysis (for the last 17 years). In contrast, salinity trends are substantially dependent on the data interval. For example, salinity trends are significantly positive along the eastern section of the transect for the data intervals between 1979 and 2000s. However, the trends became insignificant when series

are extended series to the 2010s. Trends contrast even more for stations off the Po River mouth, where surface trends switch from negative to positive and then to negative with changing interval (1979 to early 2000s vs. 1979 to mid and late 2000s vs. 1979 to 2010s). Therefore, interannual to decadal variability of the Po River freshwater outflow may substantially alter the salinity trends in the northern Adriatic. That also refers to the PDA trends, sensitivity of which to the length of the examined





time series is dominantly influenced by salinity changes in the surface layer. Yet, temperature effect keeps PDA trends more stable and mostly significantly negative in the bottom layer.

However, an uneven sampling though a year was conducted over the time interval, with more data gaps present in the first 11 years of monitoring (1979-1989). This particularly applies to autumn, when the maximum in temperature is present in near-bottom layers (30 m, Fig. 4). To test if these gaps might affect the annual trend computation, we mirrored the gaps to the last 11 years of measurements (2007-2017) and then estimated the trends. Precisely, the data gap present in 1979 at a certain station, depth and parameter is imposed to 2017 through omitting of the respective data, the gaps in 1980 are imposed to 2016, etc. Then, the annual trends are recomputed (not shown). The temperature trend shows a slight decrease, up to 10% at the most of the transect, and larger decrease (about 20 %) at 30 m of stations SJ105 and SJ108. Yet, overall distribution of trends and of their significance over the transect remained the same. Salinity trends were not affected much, while the PDA trends resemble a small but not significant decrease over the transect driven by the decrease in temperature trends.

## 4 Discussion and conclusions

The study reveals several important conclusions coming from the presented analysis of multidecadal temperature, salinity and density dataset measured in the northern Adriatic between 1979 and 2017:

1. Albeit the northern Adriatic is quite shallow, its vertical stratification is persistent through most of the year (March-November) due to seasonal heating and freshening by a substantial river inflow.
2. Interannual to decadal changes in bottom temperature and salinity are not correlated, acting on different time scales and indicating different dominant mechanisms governing their variability.
3. Temperature trends are strongly positive, significant and persistent in time all over the investigated region, slightly overestimated by a change in sampling strategy over seasons, larger near the surface and during spring and summer.
4. Salinity trends are weak and mostly insignificant, changing with data interval used for the trend computation, whilst reflecting changes in the plume dynamics of the major Adriatic river, Po River, and driven by changes in stratification.
5. From surface salinity trends at stations SJ101 and SJ108, it can be hypothesized that the transport of freshwater has been reduced towards the southeast along the coast, while the fresh waters were more transported and kept on the northern Adriatic shelf.
6. Wintertime PDA trends indicate that lighter dense waters have been generated in the northern Adriatic during recent decades, dominantly due to temperature changes.

The observed temperature trends are slightly higher than the average sea surface temperature trends in the Mediterranean in the examined period, which are also found to peak in summer (Shaltout and Omsted, 2014; Pastor et al., 2018). The largest Mediterranean-wide trend derived from satellites (1982-2016, Pastor et al., 2018) is observed in June (4.3°C over 100 years), while being even larger in the northern Adriatic (ca. 5.5°C over 100 years). The surface trend estimates in June derived from *in situ* measurements at the Rovinj-Po transect (1979-2016) are 6.3-8.5°C (exception is SJ108, which is





behaving differently as it is strongly influenced by the Po River plume). The lowest satellite-derived trends in the northern Adriatic are determined for October and equal to approximately 0.6ºC over 100 years, versus 1.5-3.0ºC over 100 years observed at the northern Adriatic transect. Overall, both satellite-derived and *in situ*-derived sea surface temperatures have the maximum in summer (June-July), while being the lowest in October and January-February. Such large temperature trends in

the northern Adriatic and in the examined period might be (i) due to much shallower thermocline in the northern Adriatic than in the rest of the Adriatic and Mediterranean (Franco et al., 1992; Artegiani et al., 1997; Lipizer et al., 2014), resulting in a larger accumulation of heat energy near the surface, and (ii) a reflection of variability at larger spatial and temporal scale, as sea surface temperature had a negative trend in the coastal eastern Adriatic between 1960 and 1975, while this trend was strongly positive between 1979 and 2015 (2.3-3.2ºC over 100 years, Grbec et al., 2018). For that reason, *in situ* sea surface

temperatures trends obtained over the middle Adriatic transversal transect between 1952 and 2010 (Vilibić et al., 2013) were found much lower, about 1.0ºC over 100 years along the eastern section of the transect. Yet, these trends were much stronger (about 2.5ºC over 100 years) along the western side of the middle Adriatic transect. Regardless of decadal and multidecadal oscillations, like the Atlantic Multidecadal Oscillation (Knight et al., 2006), which is found responsible for a half of the sea surface temperature trend in the Mediterranean (Marullo et al., 2011; Skliris et al., 2012; Macias et al., 2013), the temperature

trends in the whole Mediterranean have been positive during the 1950-2015 period (Iona et al., 2018). The northern Adriatic trends follow these findings.

Differently, the Adriatic salinity fluctuations have been largely affected by the Adriatic-Ionian Bimodal Oscillating System (BiOS, Gačić et al., 2010), which – on a decadal timescale – alters salinity in the Adriatic through advection of the highly saline LIW from the Levantine Basin or less saline Western Mediterranean waters. These alterations have been

documented to affect middle and southern Adriatic (Gačić et al., 2010; Mihanović et al., 2015), yet not proven to affect the shallow northern Adriatic, which is still considered to be mostly affected by local processes, including a substantial freshwater load by rivers (Franco et al., 1992; Artegiani et al., 1997). Recent investigations on some chemical parameters (Dautović et al., 2017) and bivalve growth (Peharda et al., 2018) along the eastern coast of the northern Adriatic document a qualitative matching and significant correlations between these variables and the BiOS regimes. Our results are supportive to that

hypothesis, as prolonged (5-10 years) periods of increased or lowered salinity can be found in the northern Adriatic, lagging after the BiOS reversals in the northern Ionian Sea for a few years. In order to properly quantify the effects of different local and remote drivers to the northern Adriatic thermohaline variability, further research is needed.

Still, salinity trends in the northern Adriatic are not significant over the investigated period, which differs from them from trends obtained in the middle Adriatic (Vilibić et al., 2013) or the whole Adriatic (Lipizer et al., 2014) over a longer time

interval. Further, salinity increase in the Adriatic is found largest of all Mediterranean basins in both present climate investigations and climate projections (Somot et al., 2006; Iona et al., 2018), whilst the whole Mediterranean is projected to salt (Vargas-Yanez et al., 2017). Whilst no significant trend in annual freshwater discharge has been documented in recent decades in the northern Adriatic (Zanchettin et al., 2008; Montanari, 2012), the difference in salinity trends between the middle and northern Adriatic reveals a change in the freshwater dynamics in the northern Adriatic. Conclusively, increased





stratification in the northern Adriatic and intensified spreading of river waters over the shallow shelf due to increased thermally-driven stratification, might be one of processes responsible for weakening of the Western Adriatic Current and the thermohaline circulation (Vilibić et al., 2013).

This study emphasizes the importance of continuous *in situ* monitoring of thermohaline and other ocean parameters over a prescribed set of stations and with satisfactorily resolution for climate studies (e.g. She et al., 2016). Resolution of measurements used in our analysis, is monthly to bimonthly. These observations are scarce in the Mediterranean but continuous at two transect in the Adriatic Sea. The maintenance of these observations is a key for proper assessment of climate changes, which might be rapid in coastal areas and might have wider consequences, such as weakening of thermohaline circulation, deoxygenation of deep regions, changing of the biogeochemical properties and fluxes and, at end, impacting the living organisms and fisheries of a region (Tintore et al., 2013).

**Acknowledgements:** We appreciate the work of researchers, technicians and research vessel crew of the Center for Marine Research at the Ruđer Bošković Institute in collecting the temperature and salinity data used in the study. The research is supported by the Croatian Science Foundation through the ADIOS project (HRZZ grant IP-06-2016-1955).



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



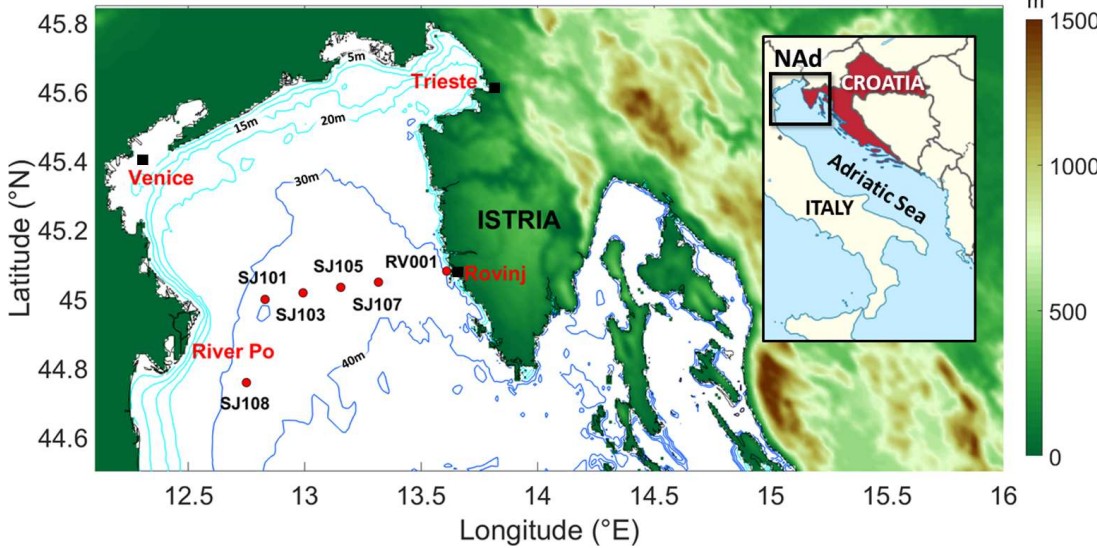

Figure 1. The northern Adriatic orography and bathymetry and location of stations.





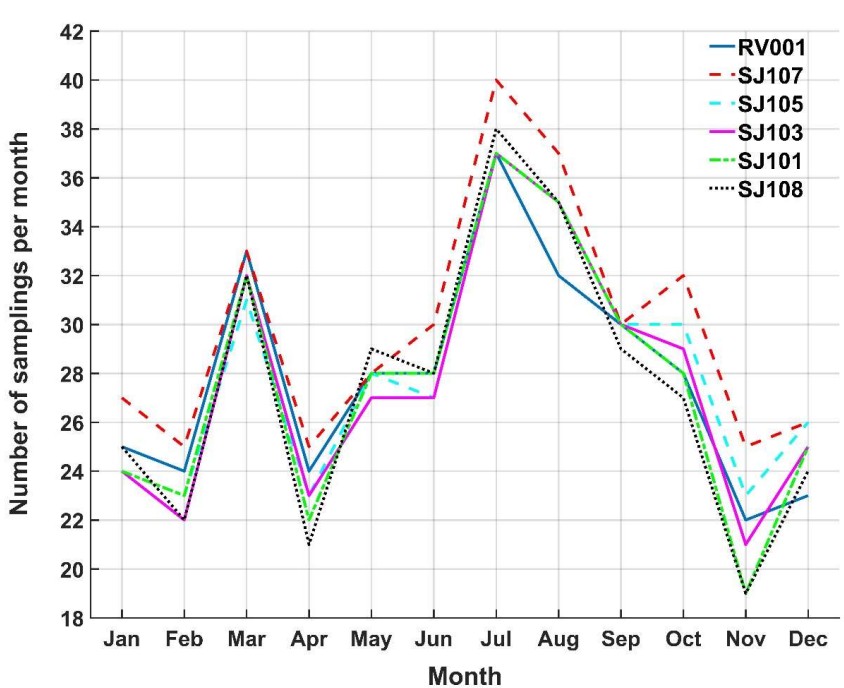

Figure 2. Total number of samplings per month during the studied period.



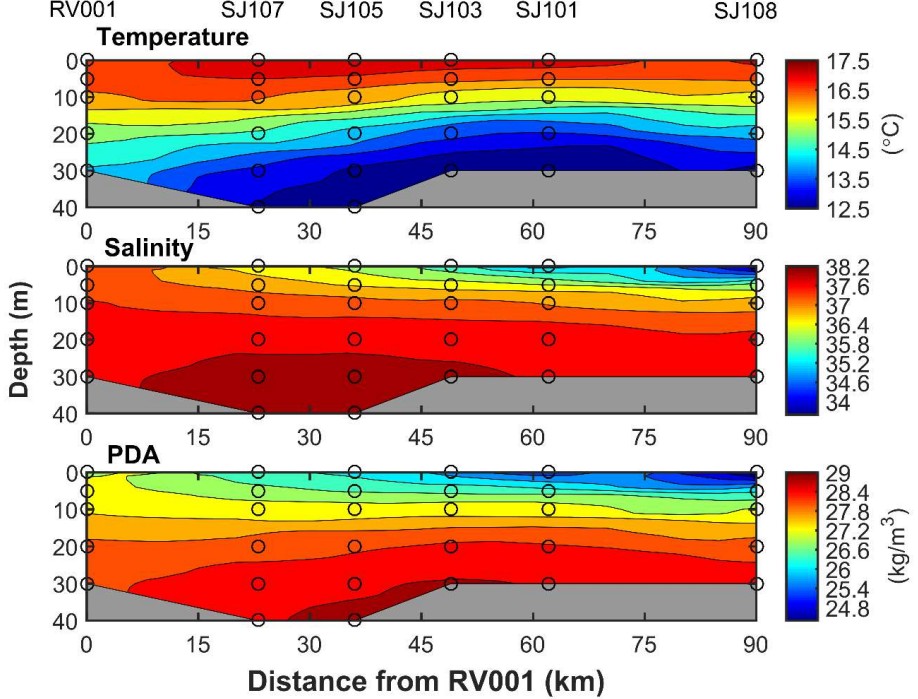

**Figure 3. Mean temperature, salinity and PDA values, estimated from residual series, across the Rovinj-Po transect.**





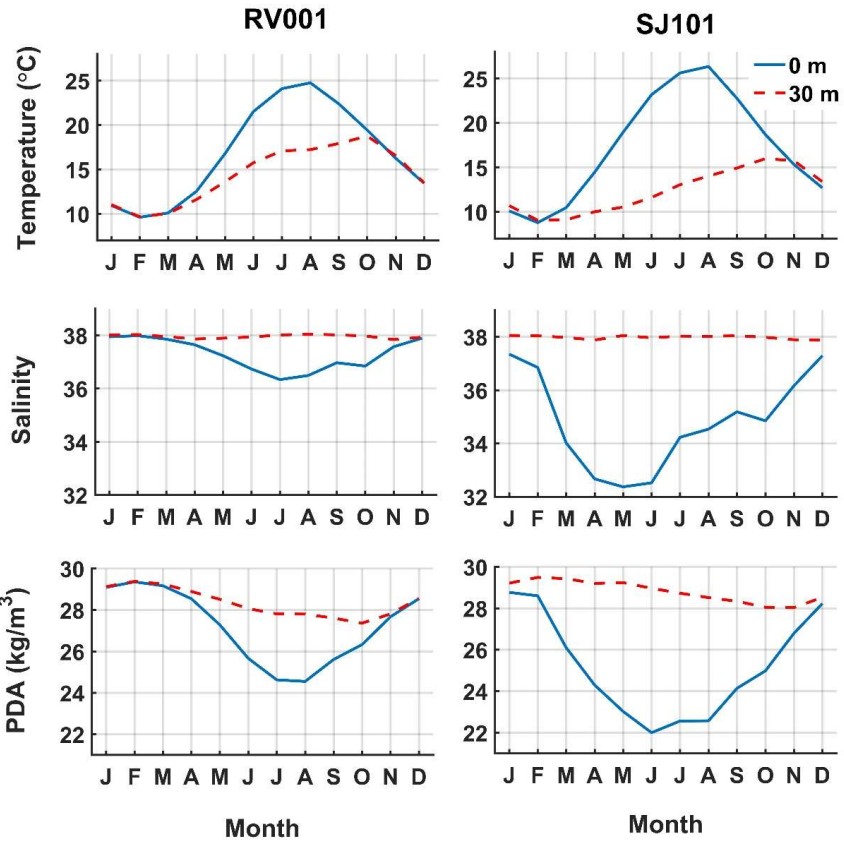

**Figure 4. Seasonal course of temperature, salinity and PDA at RV001 (left) and SJ101 (right).**




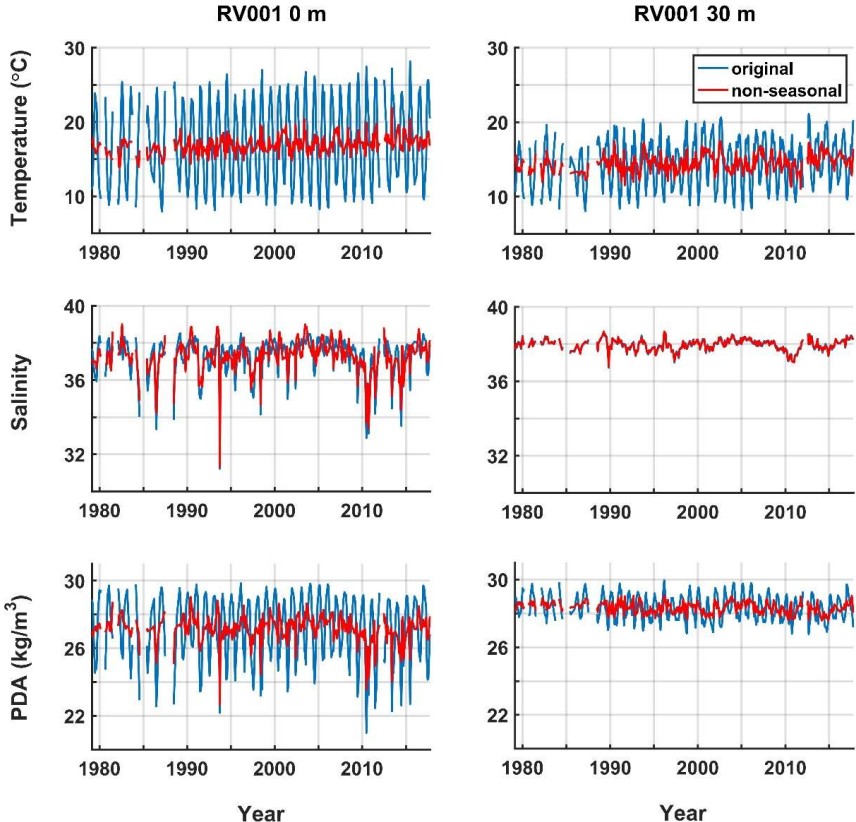

Figure 5. Original and residual components of the temperature, salinity and PDA series at surface (0 m, left) and bottom (30 m, right) of RV001.



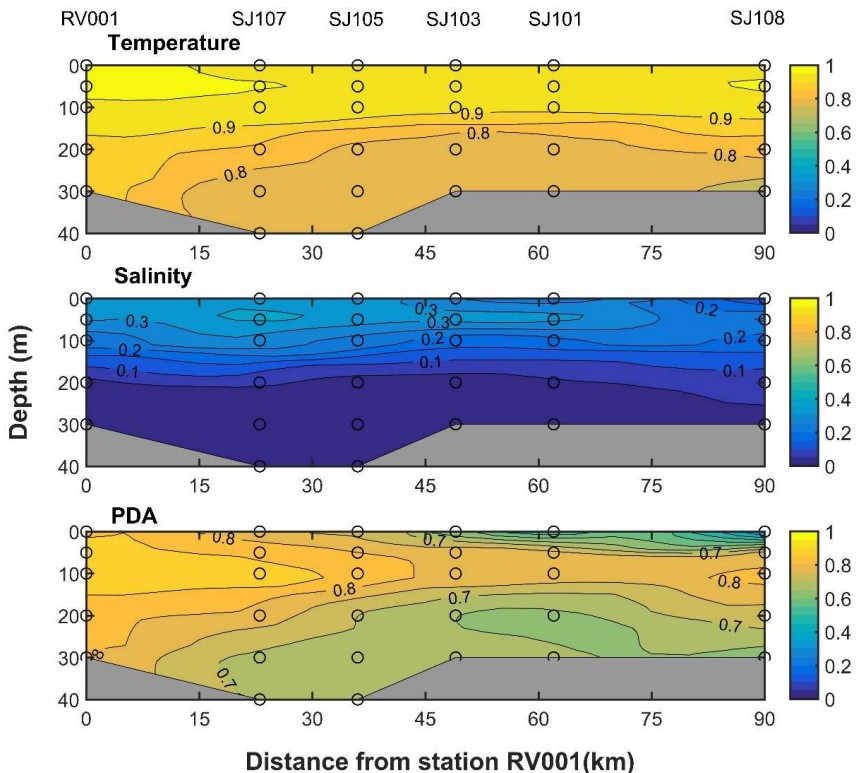

Figure 6. Ratio between seasonal and total variance of temperature, salinity and PDA.



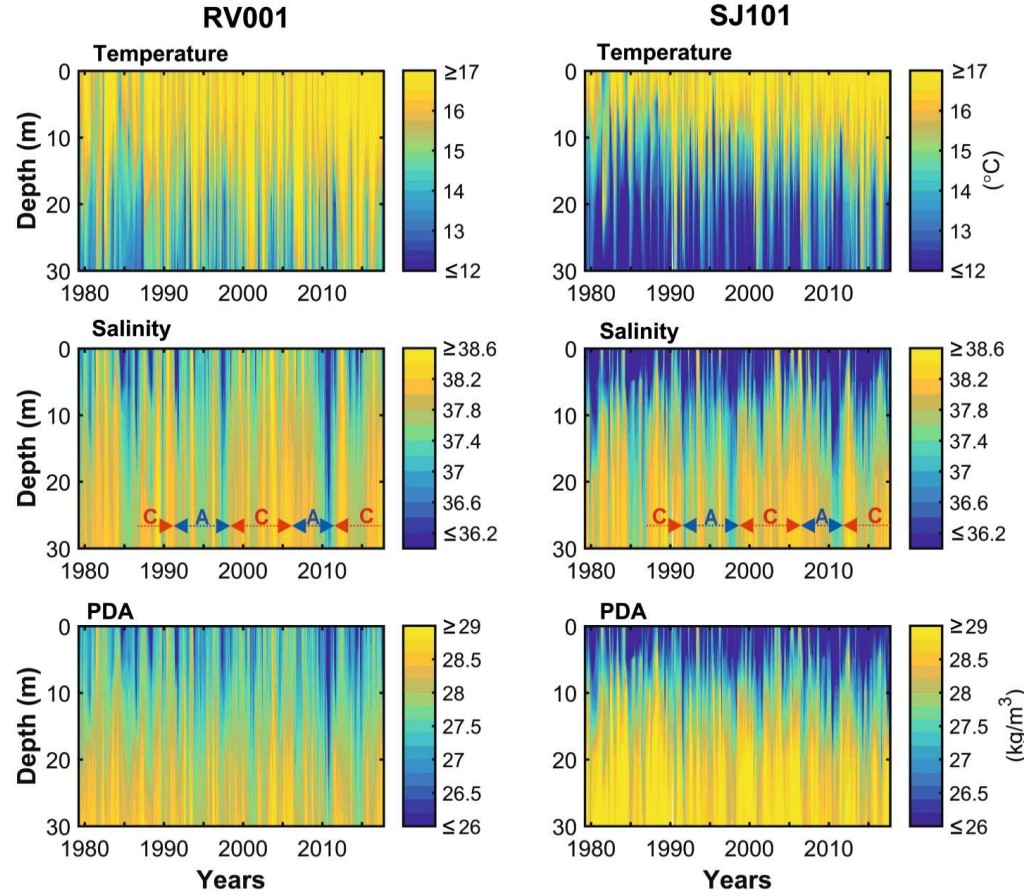

**Figure 7. Hovmoller plots of temperature, salinity and PDA at RV001 and SJ101. Seasonal cycle has been removed. BiOS regimes**
5  **in the northern Ionian Sea have been marked (A - anticyclonic, C - cyclonic) following Mihanović et al. (2015).**

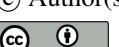


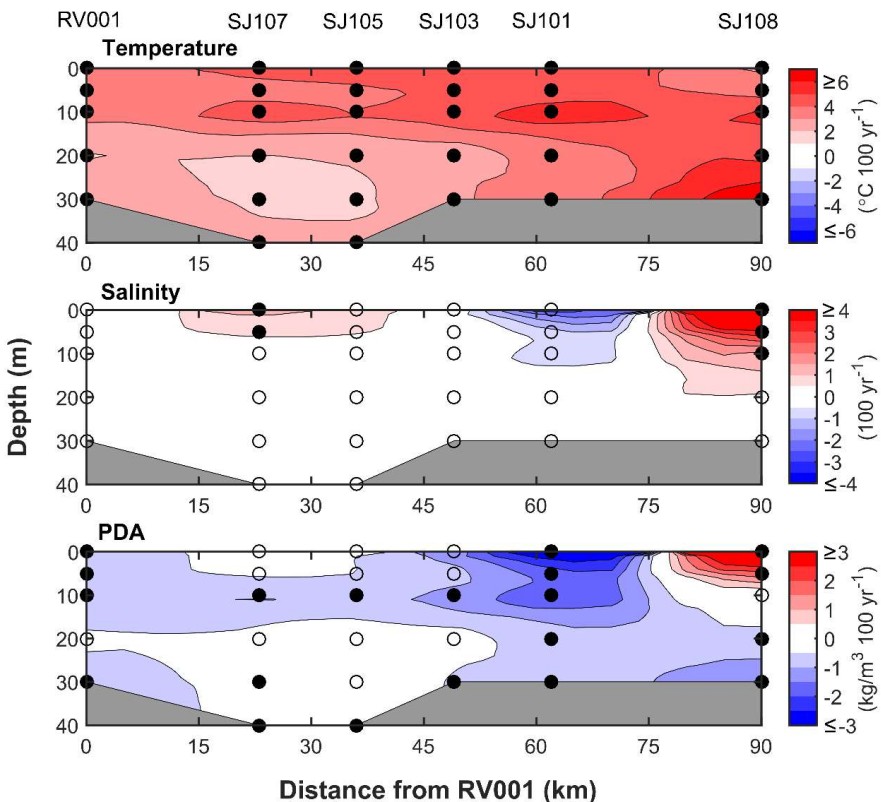

**Figure 8. Annual trends in residual temperature, salinity and PDA. Filled circles denote trends significant at 95%.**





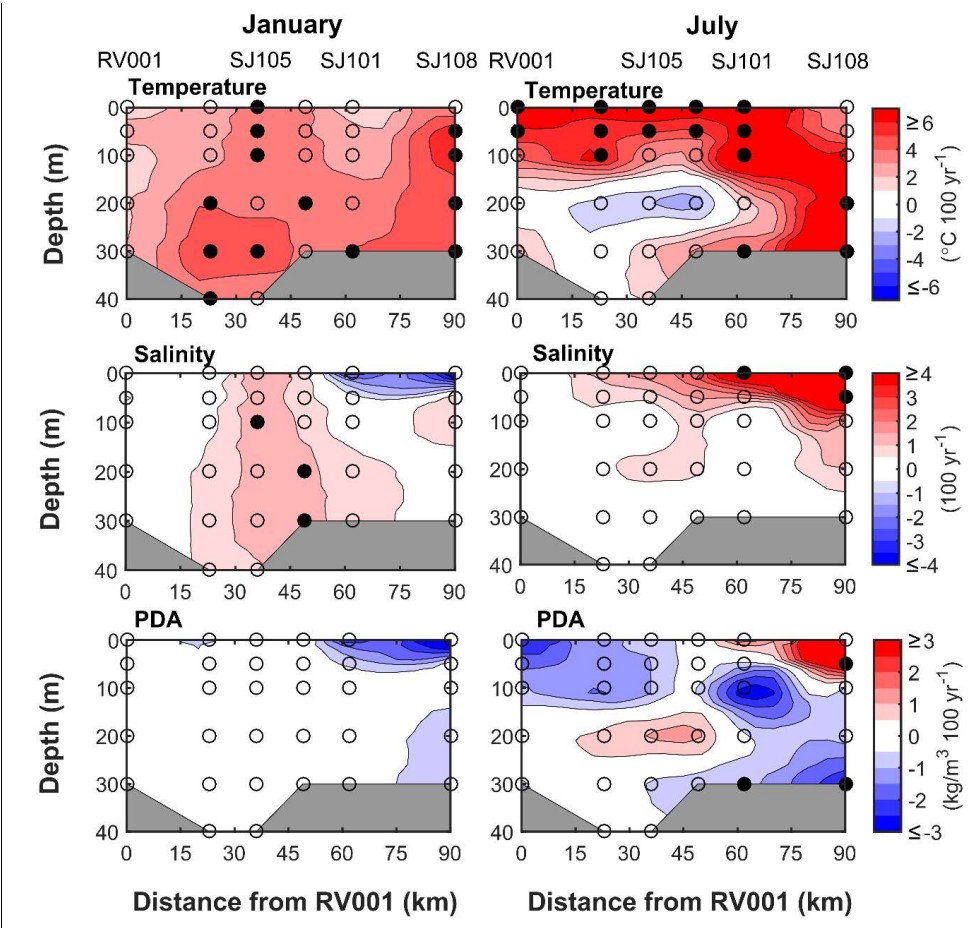

**Figure 9. January and July trends in residual temperature, salinity and PDA. Filled circles denote trends significant at 95%.**





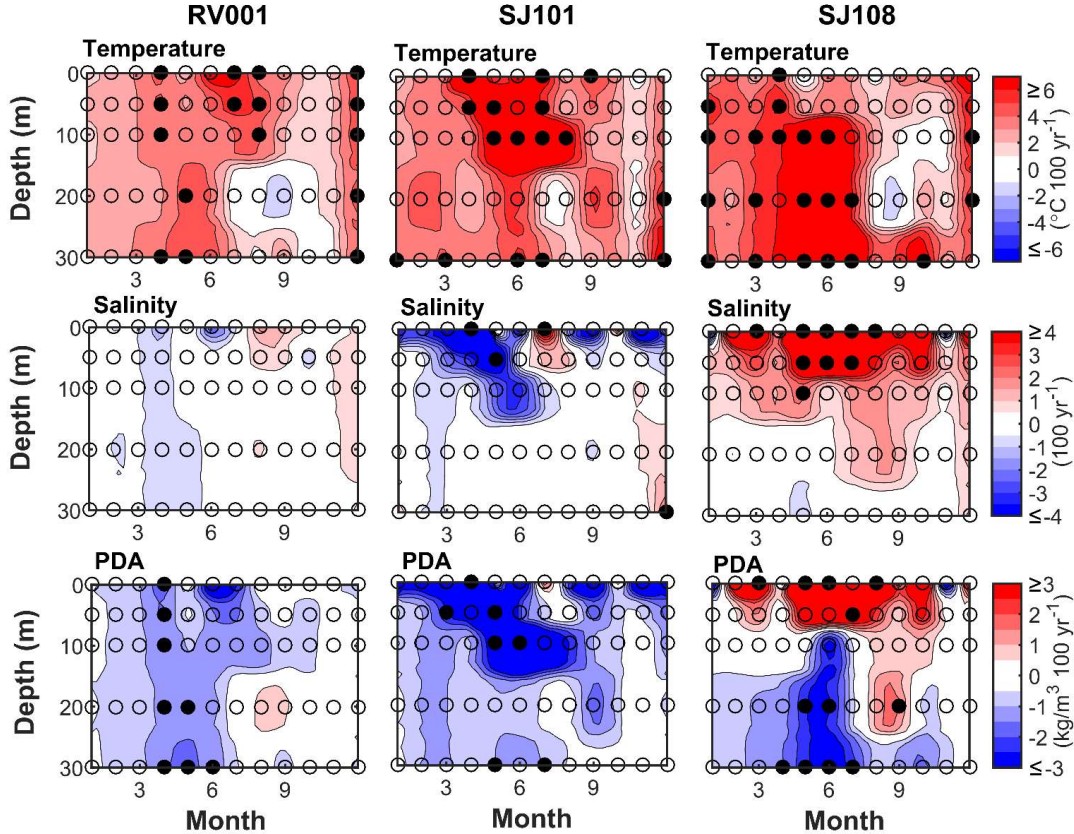

**Figure 10. Distribution of monthly trends in residual temperature, salinity and PDA over a year at RV001, SJ101, and SJ108. Filled circles denote trends significant at 95%.**



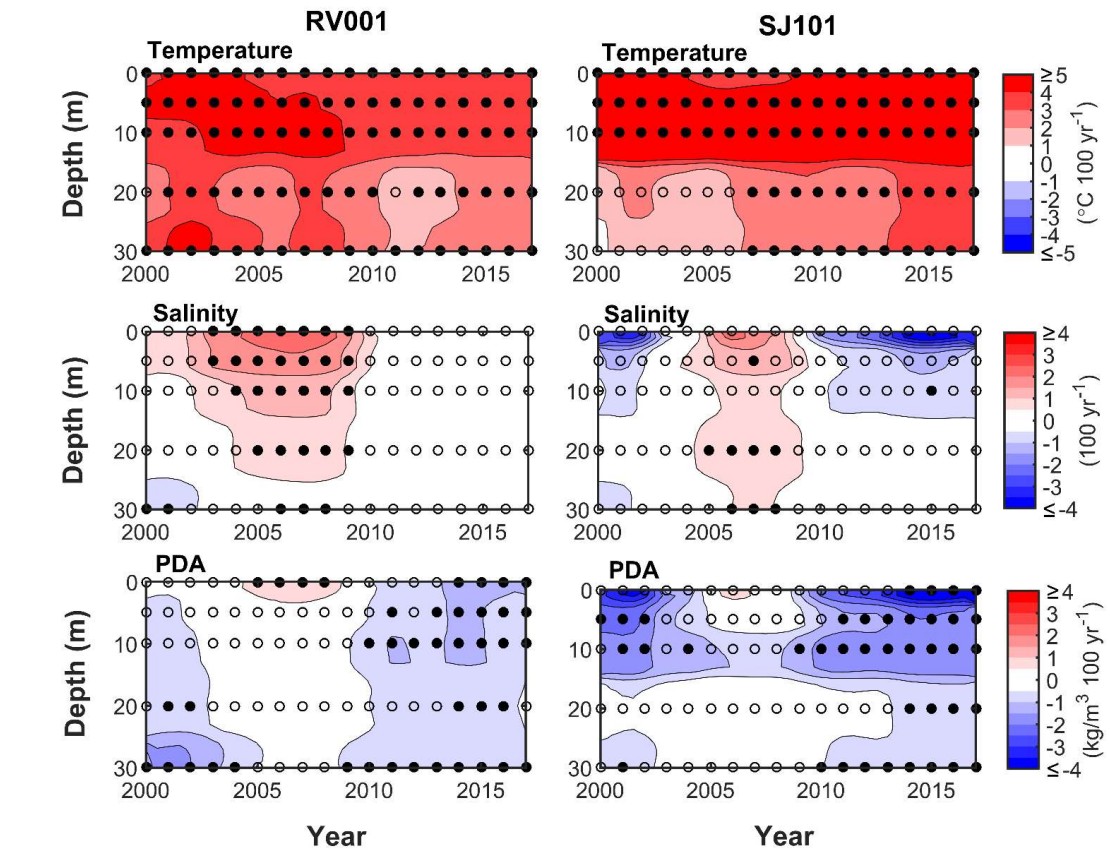

Figure 11. Sensitivity of trend estimates to the length of the series at RV001 and SJ101 stations. The trends are estimated between 1979 and the year indicated at the x-axis. Filled circles denote trends significant at 95%.

