# Peer review of "Present-climate trends and variability in thermohaline properties of the northern Adriatic shelf"

_Ocean Science, 2019_

## Referee Comment (RC1)

Review of the paper //doi.org/10.5194/os-2019-10:

Present-climate trends and variability in thermohaline properties of the northern Adriatic shelf
By: Vilbic, I.,Zemunik, P. Šepić J., Dunić, N., Marzouk, O.,Mihanović, H., Denamiel, C., Precali, R. and Djakovac, T.

The paper describes the analysis of a 38 year long time series in the northern Adriatic Sea which consists of 6 stations extending from the Croatian to the Italian coastline. The analysis examines seasonal to decadal variability and trends of the hydrographic parameters, temperature, salinity and density. The data and analysis is described well in the paper and the discussion and the conclusions can be well understood from the described analyses and the figures. The northern Adriatic is only a small sea area but it is extremely important for the Mediterranean Sea as a supplier of the dense bottom water AdDW. The discussion about long term changes (variability) of the deep water characteristics coming from the northern Adriatic is however only short. Nevertheless, I consider the paper to be relevant to Mediterranean research in this sense. The further results achieved are not new in the narrower sense (the authors have already published a number of publications on the time series here), but a consistent continuation of their work. Changes of decadal time scales require long observation periods, therefore I consider the continued analysis of this time series to be scientifically relevant. However, there remain some inaccuracies and questions about the text. These are specified below. All in all, I consider the paper to be suitable for publication after major revisions.

Here some more detailed comments, questions and corrections:

1: In the abstract I would avoid making innuendos like "indicating different mechanisms which govern their variability" (which?, line 13) or "indicate substantial changes in the thermohaline circulation" (which, line 17).

2: page 3, line 7: if you use abbreviations, please define them beforehand (ERA)

3: page 4, line 4: a winter, change just to winter

4.: page 4, line 5 and 6: through "the" rest of the year / when "the" thermocline is / increasing again "the" stability

5: page 4, line 8: what means in this context "vertical mixing prevails to the buoyancy"? (stratification is less?)

6: page 4, line 32: what are the "overall temperature changes"? Changes in the original time series?

7: page 5, line 4: "the" variance of "the" salinity seasonal series

8: page 5, line 5/6: "as well transient changes occurring over a few month" how can they dominate in the series? I thought, they were filtered out?

9: page 5, line 11: there is a discrepancy between fig. 7 and the text (SJ101 and SJ108?), so I can therefore not really understand what is said.

10: page 5, line 13: change to: bottom layer everywhere except at station RV001

11: page 5, line 21 and 25: "the" residual salinity series / to affect "the" southern and middle …

12: page 5, line 28: are lagging a few years: Did you investigate this further? How many years? How big is the range of lag?

13: page 5, line 31/32: I don't get what is said in brackets and why do I know that interannual to decadal changes are dominantly affected by salinity?

14: page 6, line 8: "indicating a weakening of stratification" why? The gradient could have been remained.

15: page 6, line 18/19: I can't follow the example (i.e. …) from the text or the figures.

16: page 6 line 24, 26, 28 and 31: "the" central part … and at station SJ108 / "The" salinity trend follows "the" temperature trend / "have more complex spatial structure" / "the" central and eastern parts of the transect / "The salinity trend in July"

17: page 7, line 1-10: I really do not understand this! How can you discuss seasonal changes from your residual time series? Seasonal changes were removed, weren't they? Was the filter not effective?

18: page 7, line 11: differ more OR differ a lot

19: page 7, line 15-17: I don't understand this conclusion from what is said before.

20: page 7, line 23-24: Out of context. What's that sentence supposed to say to me here?

21: page 8, 3-11: Why are the gaps mirrored and not chosen arbitrary to see the effect of gaps in the analysis?

22: page 8, conclusion 2: very general, which different dominant mechanisms are meant?

23: page 8, conclusion 5: what does this mean for the circulation of the northern Adriatic?

24: page 8, line 28: "Our" observed temperature trends (because otherwise it is misleading)

25: page 9, line 7-16: what is meant by reflection of variability? The conclusion ii) confuses me, I don't really get what the authors want to say. Please reformulate.

26: page 9, line 20: "the" middle and

27: page 9, line 28: "which differs from trend" (skip from them)

28: page 10, line 2: one of "the" processes for weakening the Western …

29: figure 3, 6, 8 and 9: Please rotate the x-axes. West should be on the left and east on the right side as usual

30: figure 7: Hardly to distinguish the different periods. Perhaps you can enlarge the figure.

---

## Referee Comment (RC2) · Anonymous Referee #2 · 2 Apr 2019

General Comments:

In this study, the authors present the thermohaline properties derived from six shallow stations in the northern Adriatic shelf, over the last four decades. The study focuses mainly on the trends and on the interannual to decadal variability of the water column structure, across an observational network spanning the area from the Croatian to the Italian coast.

The study follows previous published work from the authors, to the point that it is hard for the reader to follow what is new in this work. I would advise the authors to make more clear the added value and the new components of this work, instead of providing a long list of references in the Introduction section. Other than that, the manuscript is clear, concise and well written, and provides a detailed description of the thermohaline

properties of the region and the changes observed over the years. This is a dense water formation area and therefore an important area for the thermohaline properties and the circulation of the eastern Mediterranean. I have found that the analysis of results for the interannual variability and the trends is valid though too descriptive, and what is missing from the manuscript is a proper attribution to physics governing the dynamics of the area. In most cases this attribution is very general and in some cases the assumptions are far-fetched.

Overall, I find the manuscript worthy of publication in Ocean Science, after a major revision. Please find below a list of comments, that I would like the authors to address in the revised manuscript.

Specific comments:

1) The Figs. 3, 6, 8, 9 appearing as "Distance from RV001 (km)" are confusing in terms of east-west direction and should be reversed in the x-axis.

2) In the Data and Methods section the authors discuss that an annual and semi-annual filter is applied to a monthly/bimonthly timeseries, producing a residual timeseries that in my understanding does not have seasonal variability. Yet, later in the text they discuss the seasonal cycle (e.g. section 3.1, Fig. 4) and present monthly trends (e.g. section 3.3, Figs. 9, 10). How is this possible, what do I have missed here? Clarify in the text.

3) Page 5, line 13-14: "...indicating the dominance of...and salinity variability". This is a good example of what I mean when I say the authors give very general explanations regarding the dynamics of the area. Can you make a fair assumption why is this observed? Clarify in the text or remove such expressions.

4) Page 5, line 15: "...hemispheric patterns...". You mean teleconnections? Is this part of an explanation to the previous phrase? Clarify in the text. 5) Page 5, line 28-29: "It looks like...BiOS reversals". The authors provide no such proofs in the text between

the link of salinity and BiOS (also elsewhere in the text). This is not a naturally pertinent argument derived from the findings of this work. The authors should remove the BiOS assumptions from the Results section and from Fig. 7, since there are not results of their work. Providing a reference in the Result section (as in page 6 line 2) does not change the fact that this is a rather forced claim here. Near the end of Section 4, the authors discuss again the possible influence of the regions dynamics with respect to BiOS. I guess in the Discussion and Conclusion section this is relevant, as long as it is clearly stated that it is not proven yet and further research is needed (which is indeed the case in the text).

6) Section 3.3 discussing the thermohaline trends is very carefully written and it is a nice addition to the manuscript, with the only exception the monthly trends that need to be clarified (see comment 2).

7) In the beginning of Section 4, there are numbered conclusions, which although seem logical and valid, some of them are a bit vague (e.g. page 8, lines 17-18"...acting on...their variability"). Can the authors be more specific?

Best regards.

---

## Author Comment (AC1) · 6 May 2019

We thank the reviewer for very careful review and constructive comments, which will be used in improvement of the manuscript. As requested, we are addressing all raised comments and suggestions, as follows:

—–

The paper describes the analysis of a 38 year long time series in the northern Adriatic Sea which consists of 6 stations extending from the Croatian to the Italian coastline. The analysis examines seasonal to decadal variability and trends of the hydrographic parameters, temperature, salinity and density. The data and analysis is described well in the paper and the discussion and the conclusions can be well understood from the

described analyses and the figures. The northern Adriatic is only a small sea area but it is extremely important for the Mediterranean Sea as a supplier of the dense bottom water AdDW. The discussion about long term changes (variability) of the deep water characteristics coming from the northern Adriatic is however only short. Nevertheless, I consider the paper to be relevant to Mediterranean research in this sense. The further results achieved are not new in the narrower sense (the authors have already published a number of publications on the time series here), but a consistent continuation of their work. Changes of decadal time scales require long observation periods, therefore I consider the continued analysis of this time series to be scientifically relevant. However, there remain some inaccuracies and questions about the text. These are specified below. All in all, I consider the paper to be suitable for publication after major revisions.

- Thanks for nice words, we are going to broaden the discussion and make some statements more precise as suggested.

Here some more detailed comments, questions and corrections: 1: In the abstract I would avoid making innuendos like "indicating different mechanisms which govern their variability" (which?, line 13) or "indicate substantial changes in the thermohaline circulation" (which, line 17).

- We will make the abstract more concise and avoid imprecise statements.

2: page 3, line 7: if you use abbreviations, please define them beforehand (ERA)

- Ok, to be done.

3: page 4, line 4: a winter, change just to winter

- To be changed.

4.: page 4, line 5 and 6: through "the" rest of the year / when "the" thermocline is / increasing again "the" stability

- Sorry, to be corrected.

[Figure]

5: page 4, line 8: what means in this context "vertical mixing prevails to the buoyancy"? (stratification is less?)

- We will rephrase this sentence.

6: page 4, line 32: what are the "overall temperature changes"? Changes in the original time series?

- Yes. We will delete the word "overall".

7: page 5, line 4: "the" variance of "the" salinity seasonal series

- To be corrected.

8: page 5, line 5/6: "as well transient changes occurring over a few month" how can they dominate in the series? I thought, they were filtered out?

- We didn't apply a low-pass filtering to the series, just filtered out annual and semi-annual cycles which are reflecting seasonal changes and are dominant in some variables. Intra-annual variations are still present in the series.

9: page 5, line 11: there is a discrepancy between fig. 7 and the text (SJ101 and SJ108?), so I can therefore not really understand what is said.

- The text is referring not just to Fig. 7, but to all stations among which SJ108 is the only one with no significant correlations between T and S at surface (0 m). We will rewrite the text and clarify this issue.

10: page 5, line 13: change to: bottom layer everywhere except at station RV001

- To be changed.

11: page 5, line 21 and 25: "the" residual salinity series / to affect "the" southern and middle . . .

- To be corrected.

12: page 5, line 28: are lagging a few years: Did you investigate this further? How many years? How big is the range of lag?

- The lag is 2 to 4 years between BiOS reversal and a change of salinity in the northern Adriatic, what is achieved by lagged correlation analyses. Yet, this finding we plan to publish in the subsequent paper, integrally with analysis of all drivers (local – heat flux, precipitation, river discharges, ..., and remote – hemispheric indices like NAO, EA, EAWR, SCA, ...) relevant for thermohaline variability in the northern Adriatic.

13: page 5, line 31/32: I don't get what is said in brackets and why do I know that interannual to decadal changes are dominantly affected by salinity?

- The text will be clarified in revised version.

14: page 6, line 8: "indicating a weakening of stratification" why? The gradient could have been remained.

- Fig. 8 (trends) are indicating positive PDA trend at surface of SJ108, i.e. an increase in PDA values in time, exactly where low mean values are persistent mostly due to river discharges (Fig. 3). The opposite is at the bottom, where PDA values are decreasing in time. In combination, vertical gradients in PDA are decreasing in time, i.e. mean stratification is weakening. We will make the text more concise.

15: page 6, line 18/19: I can't follow the example (i.e. . . .) from the text or the figures.

- We will clarify this issue and make the text concise. If looking in positions of stations SJ101 and SJ108 (Fig. 1), salinity trend is large and positive at SJ108 (located south of the Po River delta) while negative at SJ101 (located east of the Po River delta).

16: page 6 line 24, 26, 28 and 31: "the" central part . . . and at station SJ108 / "The" salinity trend follows "the" temperature trend / "have more complex spatial structure" / "the" central and eastern parts of the transect / "The salinity trend in July"

- Sorry, to be corrected.

17: page 7, line 1-10: I really do not understand this! How can you discuss seasonal changes from your residual time series? Seasonal changes were removed, weren't they? Was the filter not effective?

- We will change the terminology, to avoid misunderstanding. Basically, we are discussing here residual trends obtained separately in different months, i.e. the trends in January, in February, ... Sorry to not putting it concisely.

18: page 7, line 11: differ more OR differ a lot

- To be corrected.

19: page 7, line 15-17: I don't understand this conclusion from what is said before.

- Ok, we will rephrase the text and make the conclusion concise.

20: page 7, line 23-24: Out of context. What's that sentence supposed to say to me here?

- We will remove the sentence.

21: page 8, 3-11: Why are the gaps mirrored and not chosen arbitrary to see the effect of gaps in the analysis?

- We choose such an approach as reflecting real problems in the data series which are already part of the calculations (at the beginning of the series). Yet, we agree that it might be done differently, as a kind of sensitivity analysis.

22: page 8, conclusion 2: very general, which different dominant mechanisms are meant?

- We will clarify this issue. Surface layers are dominantly driven by processes acting at the surface (water budget, heat budget), while changes in bottom layers are more reflecting processes advecting water masses from the southeast to the northern Adriatic.

23: page 8, conclusion 5: what does this mean for the circulation of the northern

Adriatic?

- That is a good question. We will add some discussion on that in the manuscript.

24: page 8, line 28: "Our" observed temperature trends (because otherwise it is misleading)

- Ok, we will change the text accordingly.

25: page 9, line 7-16: what is meant by reflection of variability? The conclusion ii) confuses me, I don't really get what the authors want to say. Please reformulate.

- The statement will be reformulated.

26: page 9, line 20: "the" middle and

- To be corrected.

27: page 9, line 28: "which differs from trend" (skip from them)

- To be corrected.

28: page 10, line 2: one of "the" processes for weakening the Western . . .

- To be corrected.

29: figure 3, 6, 8 and 9: Please rotate the x-axes. West should be on the left and east on the right side as usual

- We will change orientation of these figures in the revised manuscript.

30: figure 7: Hardly to distinguish the different periods. Perhaps you can enlarge the figure.

- Following suggestion by Reviewer #2, we will remove the periods from the figure as relations between the BiOS and thermohaline oscillations are not sufficiently proven in this manuscript.

---

## Author Comment (AC2) · 6 May 2019

We thank the reviewer for very careful review and constructive comments, which will be used in improvement of the manuscript. As requested, we are addressing all raised comments and suggestions, as follows:

——

In this study, the authors present the thermohaline properties derived from six shallow stations in the northern Adriatic shelf, over the last four decades. The study focuses mainly on the trends and on the interannual to decadal variability of the water column structure, across an observational network spanning the area from the Croatian to the Italian coast. The study follows previous published work from the authors, to the point

that it is hard for the reader to follow what is new in this work. I would advise the authors to make more clear the added value and the new components of this work, instead of providing a long list of references in the Introduction section. Other than that, the manuscript is clear, concise and well written, and provides a detailed description of the thermohaline properties of the region and the changes observed over the years. This is a dense water formation area and therefore an important area for the thermohaline properties and the circulation of the eastern Mediterranean. I have found that the analysis of results for the interannual variability and the trends is valid though too descriptive, and what is missing from the manuscript is a proper attribution to physics governing the dynamics of the area. In most cases this attribution is very general and in some cases the assumptions are far-fetched.

Overall, I find the manuscript worthy of publication in Ocean Science, after a major revision. Please find below a list of comments, that I would like the authors to address in the revised manuscript.

- Thanks for your comments, we will change the manuscript accordingly.

Specific comments: 1) The Figs. 3, 6, 8, 9 appearing as "Distance from RV001 (km)" are confusing in terms of east-west direction and should be reversed in the x-axis.

- We will change orientation of these figures in the revised manuscript.

2) In the Data and Methods section the authors discuss that an annual and semi-annual filter is applied to a monthly/bimonthly timeseries, producing a residual timeseries that in my understanding does not have seasonal variability. Yet, later in the text they discuss the seasonal cycle (e.g. section 3.1, Fig. 4) and present monthly trends (e.g. section 3.3, Figs. 9, 10). How is this possible, what do I have missed here? Clarify in the text.

- Sorry for confusing the seasonal cycle and its removal in the manuscript, also as the word "seasonal" is wrongly used at some places. In current version, Figs. 4-6 are

dealing with seasonal cycle in the series: Fig. 4 shows annual courses, Fig. 5 shows the series with or without seasonal cycle (to familiarize with the methodology), while Fig. 6 presents the variance of the seasonal cycle in measured series. On the other hand, interannual variability and trend analyses (Sections 3.2 and 3.3) are performed on the residual series, the latter as making more robust the statistical significance estimates. The confusion is also probably created by putting residual mean averages at the beginning of Section 3.1 (residuals are used here as number of samples are not uniformly distributed over a year and thus the annual average might be biased).

- So, we plan to rewrite Section 3.1 in a logical manner, to start with description of the data, extraction of seasonal cycle and then computations of residual mean averages. I.e. to move the first paragraph of Section 3.1 to the end of the paragraph (with Fig. 3). More, the terminology in trend estimates will be changed regarding the use of the word "seasonal" (improperly describing the analyses), which will be changed to "trends estimated for a month".

3) Page 5, line 13-14: "...indicating the dominance of...and salinity variability". This is a good example of what I mean when I say the authors give very general explanations regarding the dynamics of the area. Can you make a fair assumption why is this observed? Clarify in the text or remove such expressions.

- We will remove this sentence, as being followed by explanations.

4) Page 5, line 15: "...hemispheric patterns...". You mean teleconnections? Is this part of an explanation to the previous phrase? Clarify in the text.

- Yes and yes. We will clarify the text.

5) Page 5, line 28-29: "It looks like...BiOS reversals". The authors provide no such proofs in the text between the link of salinity and BiOS (also elsewhere in the text). This is not a naturally pertinent argument derived from the findings of this work. The authors should remove the BiOS assumptions from the Results section and from Fig. 7,

since there are not results of their work. Providing a reference in the Result section (as in page 6 line 2) does not change the fact that this is a rather forced claim here. Near the end of Section 4, the authors discuss again the possible influence of the regions dynamics with respect to BiOS. I guess in the Discussion and Conclusion section this is relevant, as long as it is clearly stated that it is not proven yet and further research is needed (which is indeed the case in the text).

- Ok, we will remove unproven construction in relation to the BiOS, and also modify Fig. 7.

6) Section 3.3 discussing the thermohaline trends is very carefully written and it is a nice addition to the manuscript, with the only exception the monthly trends that need to be clarified (see comment 2).

- Thanks, we clarified the computation of trends and change wording.

7) In the beginning of Section 4, there are numbered conclusions, which although seem logical and valid, some of them are a bit vague (e.g. page 8, lines 17-18"...acting on...their variability"). Can the authors be more specific?

- We will rewrite vague conclusions and add specific statements.

---

## Author Response (AR1)

**Response to reviewers' comments to the manuscript "Present-climate trends and variability in thermohaline properties of the northern Adriatic shelf" submitted to Ocean Science (os-2019-10)**

We thank the reviewers for their very careful review and constructive comments, which are used in improvement of the manuscript. As requested, we addressed all raised comments and suggestions, as follows:

Reviewer #1

The paper describes the analysis of a 38 year long time series in the northern Adriatic Sea which consists of 6 stations extending from the Croatian to the Italian coastline. The analysis examines seasonal to decadal variability and trends of the hydrographic parameters, temperature, salinity and density. The data and analysis is described well in the paper and the discussion and the conclusions can be well understood from the described analyses and the figures. The northern Adriatic is only a small sea area but it is extremely important for the Mediterranean Sea as a supplier of the dense bottom water AdDW. The discussion about long term changes (variability) of the deep water characteristics coming from the northern Adriatic is however only short. Nevertheless, I consider the paper to be relevant to Mediterranean research in this sense. The further results achieved are not new in the narrower sense (the authors have already published a number of publications on the time series here), but a consistent continuation of their work. Changes of decadal time scales require long observation periods, therefore I consider the continued analysis of this time series to be scientifically relevant. However, there remain some inaccuracies and questions about the text. These are specified below. All in all, I consider the paper to be suitable for publication after major revisions.

- **Thanks for nice words, we broadened the discussion and made some statements more precise as suggested.**

Here some more detailed comments, questions and corrections:
1: In the abstract I would avoid making innuendos like "indicating different mechanisms which govern their variability" (which?, line 13) or "indicate substantial changes in the thermohaline circulation" (which, line 17).

- **We made the abstract more concise and avoided imprecise statements.**

2: page 3, line 7: if you use abbreviations, please define them beforehand (ERA)

- **Done.**

3: page 4, line 4: a winter, change just to winter

- **Changed.**

4.: page 4, line 5 and 6: through "the" rest of the year / when "the" thermocline is / increasing again "the" stability

- **Corrected.**

5: page 4, line 8: what means in this context "vertical mixing prevails to the buoyancy"? (stratification is less?)

- **This sentence is rephrased.**

6: page 4, line 32: what are the "overall temperature changes"? Changes in the original time series?

- **Yes. The word „overall" is deleted.**

7: page 5, line 4: "the" variance of "the" salinity seasonal series

- **Corrected.**

8: page 5, line 5/6: "as well transient changes occurring over a few month" how can they dominate in the series? I thought, they were filtered out?

- **We didn't apply a low-pass filtering to the series, just filtered out annual and semi-annual cycles which are dominant in some variables. Intra-annual variations are still present in the series.**

9: page 5, line 11: there is a discrepancy between fig. 7 and the text (SJ101 and SJ108?), so I can therefore not really understand what is said.

- **The text is referring not just to Fig. 7, but to all stations among which SJ108 is the only one with no significant correlations between T and S at surface (0 m). The text is rewritten and clarified.**

10: page 5, line 13: change to: bottom layer everywhere except at station RV001

- **Changed.**

11: page 5, line 21 and 25: "the" residual salinity series / to affect "the" southern and middle …

- **Corrected.**

12: page 5, line 28: are lagging a few years: Did you investigate this further? How many years? How big is the range of lag?

- **The lag is 2 to 4 years between BiOS reversal and a change of salinity in the northern Adriatic, what is achieved by lagged correlation analyses. Yet, this finding we plan to publish in the subsequent paper, integrally with analysis of all drivers (local – heat flux, precipitation, river discharges, ..., and remote – hemispheric indices like NAO, EA, EAWR, SCA, ...) relevant for thermohaline variability in the northern Adriatic.**

13: page 5, line 31/32: I don't get what is said in brackets and why do I know that interannual to decadal changes are dominantly affected by salinity?

- **If one applies a simple linear model of changing density, assuming that PDA is proportional to (alpha\*T + beta\*S), one can estimate the contribution of temperature and salinity to the PDA changes. The text in brackets is providing these estimates. We expanded and clarified the text in the revised version.**

14: page 6, line 8: "indicating a weakening of stratification" why? The gradient could have been remained.

- **Fig. 8 (trends) are indicating positive PDA trend in the surface layer of SJ108, i.e. an increase in PDA values in time, exactly where low mean values are persistent mostly due to river discharges (Fig. 3). The opposite is at the bottom, where PDA values are decreasing in time. In combination, vertical gradients in PDA are decreasing in time, i.e. mean stratification is weakening. We corrected the text and made it more concise.**

15: page 6, line 18/19: I can't follow the example (i.e. …) from the text or the figures.

- **The text is clarified and made concise. If looking in positions of stations SJ101 and SJ108 (Fig. 1), salinity trend is large and positive at SJ108 (located south of the Po River delta) while negative at SJ101 (located east of the Po River delta).**

16: page 6 line 24, 26, 28 and 31: "the" central part ... and at station SJ108 / "The" salinity trend follows "the" temperature trend / "have more complex spatial structure" / "the" central and eastern parts of the transect / "The salinity trend in July"

- **Corrected.**

17: page 7, line 1-10: I really do not understand this! How can you discuss seasonal changes from your residual time series? Seasonal changes were removed, weren't they? Was the filter not effective?

- **The terminology is changed, to avoid misunderstanding. Basically, we are here discussing residual trends obtained separately in different months, i.e. the trends in January, in February, ... .**

18: page 7, line 11: differ more OR differ a lot

- **Corrected.**

19: page 7, line 15-17: I don't understand this conclusion from what is said before.

- **The sentence is removed.**

20: page 7, line 23-24: Out of context. What's that sentence supposed to say to me here?

- **The sentence is removed.**

21: page 8, 3-11: Why are the gaps mirrored and not chosen arbitrary to see the effect of gaps in the analysis?

- **We choose such an approach as reflecting real problems in the data series which are already part of the calculations (at the beginning of the series). Yet, we agree that it might be done differently, as a kind of sensitivity analysis.**

22: page 8, conclusion 2: very general, which different dominant mechanisms are meant?

- **This issue is clarified. Temperature is dominantly driven by processes acting at the surface (heat fluxes), while changes in salinity are more reflecting processes advecting water masses from the southeast to the northern Adriatic.**

23: page 8, conclusion 5: what does this mean for the circulation of the northern Adriatic?

- **That is a good question. We believe that such change would increase eddy kinetic energy and residence time in the northernmost part of the Adriatic. We added a sentence on that in the manuscript.**

24: page 8, line 28: "Our" observed temperature trends (because otherwise it is misleading)

- **The text is changed accordingly.**

25: page 9, line 7-16: what is meant by reflection of variability? The conclusion ii) confuses me, I don't really get what the authors want to say. Please reformulate.

- **The text is reformulated.**

26: page 9, line 20: "the" middle and

- **Corrected.**

27: page 9, line 28: "which differs from trend" (skip from them)

- **Corrected.**

28: page 10, line 2: one of "the" processes for weakening the Western …

- **Corrected.**

29: figure 3, 6, 8 and 9: Please rotate the x-axes. West should be on the left and east on the right side as usual

- **The orientation of these figures is changed.**

30: figure 7: Hardly to distinguish the different periods. Perhaps you can enlarge the figure.

- **Following suggestion by Reviewer #2, we removed the periods from the figure as relations between the BiOS and thermohaline oscillations are not sufficiently proven in this manuscript.**

**Reviewer #2**

In this study, the authors present the thermohaline properties derived from six shallow stations in the northern Adriatic shelf, over the last four decades. The study focuses mainly

on the trends and on the interannual to decadal variability of the water column structure, across an observational network spanning the area from the Croatian to the Italian coast. The study follows previous published work from the authors, to the point that it is hard for the reader to follow what is new in this work. I would advise the authors to make more clear the added value and the new components of this work, instead of providing a long list of references in the Introduction section. Other than that, the manuscript is clear, concise and well written, and provides a detailed description of the thermohaline properties of the region and the changes observed over the years. This is a dense water formation area and therefore an important area for the thermohaline properties and the circulation of the eastern Mediterranean. I have found that the analysis of results for the interannual variability and the trends is valid though too descriptive, and what is missing from the manuscript is a proper attribution to physics governing the dynamics of the area. In most cases this attribution is very general and in some cases the assumptions are far-fetched.

Overall, I find the manuscript worthy of publication in Ocean Science, after a major revision. Please find below a list of comments, that I would like the authors to address in the revised manuscript.

- **Thanks for your comments, we changed the manuscript accordingly.**

Specific comments:
1) The Figs. 3, 6, 8, 9 appearing as "Distance from RV001 (km)" are confusing in terms of east-west direction and should be reversed in the x-axis.

- **We changed the orientation of these figures in the revised manuscript.**

2) In the Data and Methods section the authors discuss that an annual and semi-annual filter is applied to a monthly/bimonthly timeseries, producing a residual timeseries that in my understanding does not have seasonal variability. Yet, later in the text they discuss the seasonal cycle (e.g. section 3.1, Fig. 4) and present monthly trends (e.g. section 3.3, Figs. 9, 10). How is this possible, what do I have missed here? Clarify in the text.

- **Sorry for confusing the seasonal cycle and its removal in the manuscript, also as the word „seasonal" is wrongly used at some places. In original version, Figs. 4-6 are dealing with seasonal cycle in the series: Fig. 4 shows annual courses, Fig. 5 shows the series with or without seasonal cycle (to familiarize with the methodology), while Fig. 6 presents the variance of the seasonal cycle in measured series. On the other hand, interannual variability and trend analyses (Sections 3.2 and 3.3) are performed on the residual series, the latter as making more robust the statistical significance estimates. The confusion is also probably created by putting residual mean averages at the beginning of Section 3.1 (residuals are used here as number**

**of samples are not uniformly distributed over a year and thus the annual average might be biased).**

**So, we rewrote Section 3.1 in a logical manner, starting with description of the data, extraction of seasonal cycle and then computations of residual mean averages. I.e. we moved the first paragraph of Section 3.1 to the end of the paragraph (with Fig. 3), and change a bit. More, the terminology in trend estimates is changed regarding the use of the word „seasonal" (improperly describing the analyses), which will be changed to „trends estimated for a month".**

3) Page 5, line 13-14: "...indicating the dominance of...and salinity variability". This is a good example of what I mean when I say the authors give very general explanations regarding the dynamics of the area. Can you make a fair assumption why is this observed? Clarify in the text or remove such expressions.

- **We removed this sentence, as being followed by explanations.**

4) Page 5, line 15: "...hemispheric patterns...". You mean teleconnections? Is this part of an explanation to the previous phrase? Clarify in the text.

- **Yes and yes. The text is clarified.**

5) Page 5, line 28-29: "It looks like...BiOS reversals". The authors provide no such proofs in the text between the link of salinity and BiOS (also elsewhere in the text). This is not a naturally pertinent argument derived from the findings of this work. The authors should remove the BiOS assumptions from the Results section and from Fig. 7, since there are not results of their work. Providing a reference in the Result section (as in page 6 line 2) does not change the fact that this is a rather forced claim here. Near the end of Section 4, the authors discuss again the possible influence of the regions dynamics with respect to BiOS. I guess in the Discussion and Conclusion section this is relevant, as long as it is clearly stated that it is not proven yet and further research is needed (which is indeed the case in the text).

- **We removed unproven construction in relation to the BiOS, and also modified Fig. 7.**

6) Section 3.3 discussing the thermohaline trends is very carefully written and it is a nice addition to the manuscript, with the only exception the monthly trends that need to be clarified (see comment 2).

- **Thanks. We clarified the computation of trends and changed wording.**

7) In the beginning of Section 4, there are numbered conclusions, which although seem logical and valid, some of them are a bit vague (e.g. page 8, lines 17-18"...acting on...their variability"). Can the authors be more specific?

- **We rewrote a vague conclusions and add specific statements.**

[revised manuscript text omitted]

---

## Author Response (AR2)

**Response to reviewers' comments to the revised manuscript "Present-climate trends and variability in thermohaline properties of the northern Adriatic shelf" submitted to Ocean Science (os-2019-10-R1)**

**We thank the reviewers for their very careful review and constructive comments, which we have used to improve our manuscript. As requested, we addressed all raised comments and suggestions, as follows:**

**Reviewer #1**

The authors have addressed my comments in the revised manuscript (ms). Most parts of the ms are now improved. I am still confused about Section 3.3 "Trends" and in particular for the discussion of Figures 9 and 10 showing monthly trends.

The authors rephrased just a small part of the text to explain the method at the end of Section 2 "Data and Methods". I understand why (e.g. sampling errors) and how (e.g. fitting cosine functions) the residual timeseries are calculated. Note that fitting cosine functions is different than applying a low- or high-pass filter and indeed a monthly signal may still be present. However, the authors do not explain this clearly in the text, nor in their reply to the reviewers, and because of this there is still confusion of what part of the signal is eventually removed.

- **As requested by both reviewers, we provided additional explanation, details and references on the methodology used in the paper (new paragraph and two classical references). We hope that the methodology is properly explained now.**

One other thing that makes those two Figures 9 and 10 confusing is that for most stations, months and variables (especially for salinity and density) the results are not statistically significant (i.e. not-filled circles). The authors are careful in the text not to discuss those patterns, but in my view, the figures are misleading showing colored features that eventually are statistically insignificant.

- **Although we agree that no significance is found for a large number of stations/depths/parameters, particularly for salinity and PDA in Fig. 9, they contain a substantial reliable information which are supportive for our analyses and conclusions. There is a substantial amount of text (four paragraphs) describing these two figures, which are describing (as acknowledged by the reviewer) statistically significant patterns and reliable results only. Therefore, we are in favour to keep these figures and the associated text in the manuscript.**

For the reasons explained above, I recommend the authors to revise the text in Sections 2 and 3.3. In Section 2, a more analytical description of the method should be provided. In Section 3.3, I strongly recommend that Figures 9 and 10 are removed since there is no added value to present statistically insignificant results for most of the stations.

- **The methodology is described in great details now, while we kept (as explained above) Figs. 9 and 10 and the associated text in the manuscript.**

**Reviewer #2**

Major comments
Well written manuscript. However I agree with Reviewer no. 2 that the seasonal cycle should not be removed by cosine functions or that the chosen strategy is better explained. This should be corrected before publication. I recommend major revisions.

- **The methodology is described in great details now, also with adding appropriate references. See also the response to the Reviewer #1 comment on that.**

Minor comments
Affiliation no.2: omit "student at"
P1 L11: "The amplitude …"

- **Corrected.**

P1 L13, P6 L4 and many other places in 3.3 and 4: You should express temperature and salinity trends in K/decade or 1/decade instead of oC over 100 years. The period of observations is only 38 years. As the temperature curve may look like a hockey stick, a temperature change per 100 year does not make sense.

- **We changed all figures and the text to oC over 10 years.**

P1 L23: occurs
P2 L11: trend
P2 L28: centennial timescale
P3 L31: is referred
P4 L4: a minimum
P4 L5: a maximum
P4 L32: delete "of the most"

- **Corrected.**

P5 L22: "Bimodal Oscillating System (BiOS)". For non-experts of the Adriatic it would be good to add one sentence explaining the BiOS.

- **More details are added in discussion section.**

P6 L25: the salinity trend in January
P6 L29: was reduced
P8 L29: Omstedt
P9 L10: Regarding the rest of the Adriatic
P10 L6: satisfactory

- **Corrected.**

[revised manuscript text omitted]